# Sleep Bruxism in Children: Etiology, Diagnosis, and Treatment—A Literature Review

**DOI:** 10.3390/ijerph18189544

**Published:** 2021-09-10

**Authors:** Sylwia Bulanda, Danuta Ilczuk-Rypuła, Aleksandra Nitecka-Buchta, Zuzanna Nowak, Stefan Baron, Lidia Postek-Stefańska

**Affiliations:** 1Department of Pediatric Dentistry, Medical University of Silesia in Katowice, Traugutta sq. 2, 41-800 Zabrze, Poland; danuta.ilczuk@gmail.com (D.I.-R.); swr@sum.edu.pl (L.P.-S.); 2Department of Temporomandibular Disorders, Medical University of Silesia in Katowice, Traugutta sq. 2, 41-800 Zabrze, Poland; zuzannaewanowak33@gmail.com (Z.N.); sbaron@sum.edu.pl (S.B.)

**Keywords:** bruxism, children, bruxism etiology, bruxism diagnosis, bruxism treatment

## Abstract

(1) Background: Bruxism, a condition characterized by grinding and involuntary clenching of the teeth, is a risk factor for the development of masticatory dysfunction. It can occur together with sleep disturbances and may be associated with abnormal body movements, breathing difficulties, increased muscle activity, and heart rate disturbances. This disorder is becoming an important dental concern in children. (2) Methods: A literature review was carried out based on a search in PubMed and Google Scholar databases for articles on bruxism in children, published between 2014 and 2021. The etiology, diagnosis, and treatment of bruxism in children were of particular interest in the study. (3) Results: A total of 40 articles were included in the review. The analyzed studies indicated that the prevalence rates of bruxism in children vary widely from 13% to 49%. The etiology of bruxism is complex and incomprehensible, and the main diagnostic criteria for this condition in children are subjective observation, clinical history, and clinical examination. The recommended therapy for sleep bruxism in children is physiotherapy and psychotherapy. (4) Conclusions: Dentists and primary care physicians should correctly diagnose bruxism in children, educate parents, prevent potential consequences for oral health, and identify possible comorbidities. Appropriate clinical guidelines for the treatment and prophylaxis of bruxism in children should also be developed.

## 1. Introduction

Bruxism is a repetitive masticatory muscle activity [1], which is a risk factor for several serious health complications. This condition was described for the first time in the medical literature in 1907 by Maria Pietkiewicz [2]. It is characterized by clenching, tooth grinding, and/or bracing or thrusting of the mandible with circadian symptoms (such as facial tightness, head and neck pain, and insomnia) [3]. The American Academy of Sleep Medicine (AASM) has indicated bruxism as a sleep-related movement disorder [4]. Bruxism can be distinguished into two types: sleep bruxism (SB) and awake bruxism (AB). A person suffering from SB is unaware of the risk factors that can lead to abnormal tooth wear and the development of temporomandibular disorder (TMD). Recent research reports suggest that SB can also cause primary headaches [5], which is believed to be mainly regulated by the central nervous system [6] and may be associated with disturbances in the GABAergic and glutamatergic system of the brain [7]. Bruxism may occur together with sleep disturbances as well as body movements, breathing problems, increased muscle activity, and heart rate disturbances [8]. Sleep disorders comorbid with bruxism include obstructive sleep apnea, parasomnias, restless leg syndrome, mandibular myoclonus, and rapid eye movement disorders [9]. The prevalence of SB in children reported in studies varies widely. In addition, SB has been shown to be more common among children compared to adults [10], with the prevalence rates ranging from 13% to 49% [11]. A study by Sousa showed that the incidence of SB in adolescents was 22.2% and the prevalence was higher in male adolescents as indicated by multivariate analysis [12]. In another study, Manfredini et al. reported variations in the incidence of bruxism between 3.5% and 40.6%, with a decrease observed with age, regardless of gender. The differences in results between studies might be mainly related to the use of unreliable tools for the diagnosis of SB in children [13]. It should also be emphasized that bruxism can lead to pathological tooth destruction, failure of dental procedures (e.g., prosthetic reconstruction), pain in the temporomandibular joint (TMJ) and craniofacial muscles, limitation of the mandible mobility, and headaches [9].

Furthermore, a study by Alves et al. indicated that parents and caregivers have insufficient knowledge about the etiology of bruxism, especially SB, which may make it difficult for them to seek help and thus contribute to the exacerbation of bruxism and its complications in adulthood [2]. Therefore, this study aimed to investigate the causes, diagnosis, and treatment of bruxism in children. 

## 2. Materials and Methods

A literature review was performed based on a search in PubMed and Google Scholar databases for articles on SB in children and its causes, diagnosis, and treatment. The following keywords were used in the search to find relevant articles: “bruxism,” “children,” “etiology,” “diagnosis,” and “treatment” (in accordance with Medical Subject Headings). The search was filtered to include only papers published from 2014 to 2021, in both Polish and English language. A preselection of articles was made by reviewing their title and/or abstract, and those that did not meet the inclusion criteria were rejected. Two researchers then conducted an independent in-depth analysis of the remaining articles. The data extracted from the studies included study characteristics, participants, interventions, and results. 

The a priori inclusion criteria for the literature review were as follows: (1)Randomized clinical trials (RCTs) and observational studies evaluating the association between risk factors and bruxism;(2)RCTs and observational studies evaluating the diagnosis of bruxism; and(3)RCTs and observational studies examining the treatment of patients with SB.

The following were excluded from the literature review: (1)Case reports;(2)Articles published in other languages than English and;(3)Articles describing the methods used for clinical treatment in adult bruxers.

The quality of the included studies was assessed based on the adequacy of the study design to the research objective, risk of bias, reliability of results, statistical work, and quality of reporting.

## 3. Results

A total of 172 articles were found in the PubMed database and 11,700 (including 9 in Polish) in the Google Scholar database. After applying the inclusion and exclusion criteria and analyzing the abstracts, 40 articles were finally selected (Figure 1.). The review included those works presenting statistically significant results (*p* < 0.05).

## 4. Discussion

### 4.1. Etiology of Bruxism in Children

The etiology of bruxism is complex and incomprehensible [9]. Many risk factors have been shown to be associated with SB. Nevertheless, several issues regarding the etiology of bruxism influencing clinical management strategies remain unresolved. According to the World Health Organization (WHO), the multiple characteristics of bruxism increase the probability of diseases or injuries in a patient [14]. Based on the etiopathogenesis, bruxism can be classified as primary—idiopathic (no accompanying comorbidities) and secondary—iatrogenic (associated with diseases or caused by the use of specific medications) [6]. Despite the distinction between SB and AB, psychological factors appear to be the main cause of this functional disorder [6,9]. Of these, the most frequently cited is the emotional state, with stress and anxiety being considered as risk factors for TMD. Oliveira et al. showed that anxiety and distress are particularly found in patients with bruxism [9]. Personality traits with a tendency toward neuroticism are also identified as contributing to the development of SB in children [6]. In a study on children aged 6–8 years, Vanderas et al. found a relationship between the concentration of catecholamines in the urine and SB [6]. Gomes et al. observed that children aged 5–13 years with hyperkinetic disorders are characterized by a higher incidence of parasomnias, such as somnolence, bedwetting, nightmares, and fear of the dark. Similar results were reported in a study on children with attention deficit hyperactivity disorder [15]. In addition, it has been reported that children tend to release the tension accumulated during the day through chronic bruxism during sleep [16].

Anxiety in children is a common clinical occurrence in pediatric psychiatry. Its incidence in the general population is estimated at 2.5–5% and the clinical population at 10.6–24% [9]. Unlike the adult population, the symptoms of anxiety in children tend to change with developmental stages, which often makes its identification difficult [9]. Somatic anxiety is known to be associated with increased muscle tone. Studies based on polysomnography (PSG) have shown an increased incidence of SB, especially in children with tension headaches [11]. Evidence also suggests a relationship between anxiety and the occurrence of AB and SB [9,17]. Generalized anxiety disorder (GAD) and social anxiety disorder are identified as most commonly associated with bruxism [11]. Additionally, a strong correlation has been shown between SB and neuroticism [3]. Children who snore or have nightmares are found to more likely develop bruxism while they sleep [16]. Thus, certain sleep quality attributes can serve as indicators for the early diagnosis of bruxism by parents and health care professionals and thus reducing its consequences. 

Socioeconomic and cultural characteristics may also be associated with the occurrence of SB. This disorder is more commonly found among children from families with a better socioeconomic status [16,18], which may be related to the higher number of daily duties and demands by children, compared to children from a poor background.

Research into sleep physiology is also of interest as sleep disturbances, such as changes in breathing during sleep, have been comprehensively linked with headaches, sleep apnea, and hypopnea (SAHOS) and SB [6]. Ferreira et al. showed that a positive correlation existed in children between SB and SAHOS (11% of bruxers also had SAHOS) [19]. Sleep disturbances such as sound and light stimuli and decreased sleep time (≤8 h) were found to have a strong association with SB [20]. 

Khoury et al. postulated that SB may play a protective role during sleep, by maintaining airway patency or stimulating saliva flow to moisturize the oropharynx [21]. Gastroesophageal reflux also appears to be a risk factor for SB and perform a protective function by stimulating saliva flow through SB [10]. 

Considering that a direct relationship may exist between parafunctions and SB, biting of lips, nails, or pens and long-term use of pacifiers may play an important role in the genesis of SB in children. Therefore, the clinical prophylaxis of SB in children should include attempts to eliminate such behaviors [3].

Więckiewicz et al. demonstrated the genetic basis of bruxism. Their study showed that the rate of bruxism episodes was significantly higher in the HTR2A rs6313 TT homozygotes compared to the heterozygous patients. Moreover, a statistically significant correlation was found between SB and the sleep apnea index in HTR2A rs2770304 TT homozygotes [22].

It has been repeatedly proven that certain drugs and chemicals can increase the number of SB episodes. Among them, the most commonly mentioned compounds are selective serotonin reuptake inhibitors (e.g., paroxetine, fluoxetine, sertraline), selective norepinephrine reuptake inhibitors (e.g., venlafaxine), antipsychotics (e.g., haloperidol), flunarizine, amphetamines (e.g., methylphenidate), 3,4-methylphenidate (ecstasy), nicotine, and alcohol [23].

### 4.2. Diagnosis of Bruxism in Children

Diagnosis and clinical evaluation of bruxism is generally a complex process [1,10] and requires performing many tests, including subjective observations and medical history analysis, clinical examination, assessment with intraoral devices (the so-called mandibular advancement devices), recording of muscle activity, electromyography (EMG), and PSG [10]. Most of the epidemiological studies on SB have been carried out on children aged six to eleven years. Diagnosis is usually performed based on reports provided by family members, describing the characteristic sounds generated by tooth grinding while sleeping [12]. One in six children and adolescents show clinical signs of TMD [24], of which the most commonly observed are reduced mouth opening, clicking, crackling, TMJ, and muscle pain [24,25]. The AASM has suggested some diagnostic criteria for determining SB which are listed in Table 1.

The minimum criteria for the diagnosis of SB are as follows:
Tooth grinding or clenching while sleeping and;One or more of the following: -Abnormally worn teeth;-Bruxism-related sounds; and;-Mandible muscle discomfort [4].



Tooth wear (assessed clinically) is associated with an increased risk of SB in children. It is more intense in primary dentition as the primary teeth exhibit a lower degree of mineralization than permanent teeth. However, the observation of wear on the hard surfaces of tooth tissues does not confirm the clinical diagnosis of SB [16].

According to the International Consensus on the assessment of bruxism [1], SB can be graded as follows: possible SB—diagnosed based on a family relationship or self-assessment of noise or tooth grinding during sleep; probable SB—diagnosed based on a self-assessment of grinding and clinical features of bruxism during the functional examination, such as increased tooth wear and detection of antagonistic teeth at the time of examination, masticatory muscle pain, or fatigue and muscle hypertrophy. The final diagnosis of SB requires reporting of grinding of teeth and consistent clinical conditions, and confirmation by PSG, which measures the EMG activity of the masticatory muscles associated with grinding, with audio and video recordings, during the sleep examination [1]. For children, the most reliable clinical method to diagnose bruxism is the reporting of teeth grinding by parents or caregivers, However, most children sleep away from their parents, and so parents are not always aware of bruxism in their children [6]. PSG tests are considered the gold standard for the diagnosis of bruxism, but their use in population studies is still not feasible due to the high cost and need for qualified specialists [11,12,26].

A feature distinguishing patients initially diagnosed with bruxism from those who have not been diagnosed with this condition, despite showing some degrees of sleep dysfunction, is the duration and intensity of muscle contractions, which are significantly different in bruxism patients [9]. Hence, EMG assessment of muscle activity as part of the overnight PSG monitoring is considered most reliable. However, the overall variability of muscle activity remains a concern. For example, people with GAD (generalized anxiety disorder) show increased muscle activity and tension during wakefulness, and even when they sleep, which makes the detection of SB difficult [11]. A screening tool used for simple PSG is the Brux-off (Spes Medica, Genova, Italy) device, which allows recording the electrical activity of the masticatory muscles and heart rate at home. 

In the last 10 years, some interesting mobile devices have been introduced to facilitate data collection. These include handheld devices receiving combined EMG and electrocardiographic traces, which have shown increased accuracy compared to only EMG-based devices and may represent a promising diagnostic tool for SB [27,28].

As mentioned above, stress is a predictor of bruxism, which can be assessed by measuring the levels of salivary biomarkers such as cortisol and alpha-amylase (sAA) [25]. Repeated exposure to stressful situations can over-activate the hypothalamic–pituitary–adrenal axis and increase cortisol levels. Although previous studies have shown that the levels of cortisol and sAA were elevated in adults with TMD, no differences were observed in the secretion of salivary stress parameters in the case of children [25].

Apart from a number of factors that may influence subjective assessments, the current lack of verified criteria for the diagnosis of SB in children undoubtedly contributes to discrepancies in the literature. Early diagnosis and identification of risk factors are important for the inhibition of craniofacial changes, as well as for the relief of craniofacial pain, restoration of lost structures, and repair of facial lesions [29]. 

### 4.3. Treatment of Bruxism in Children 

A controversy regarding therapeutic management of SB prevails among clinicians. It is believed that between three and five years of age, the occlusal surfaces of teeth must undergo physiological wear to allow the growth and development of jaws [6]. In addition, studies have shown that the incidence of bruxism decreases from around nine to ten years of age, which supports the belief that most children with bruxism will not exhibit this activity in adolescence and adulthood [6,30]. Physicians should observe children for the signs and symptoms of bruxism such as tooth wear of the permanent teeth, headache, craniofacial pain, TMD, or reduced mouth opening. Treatment of bruxism is challenging and requires the cooperation of the physician, parents, and the child. Currently, SB is treated with physiotherapy [31,32]. Treatment methods commonly used for treating SB in pediatric patients are kinesiotherapy, massage, infrared therapy, and low-level laser therapy (LLLT) [31]. Among them, LLLT is noninvasive, cost-effective, painless, and requires a shorter exposure time per acupoint [31].

Dental treatment of bruxism involves the use of occlusal appliances during sleep in order to protect the teeth against pathological abrasion (Table 2). Reports also indicate that orthodontic procedures aimed at widening the jaw are performed to reduce the incidence of SB in children [33]. Limited data show that treatment with occlusal appliances is effective in the case of deciduous or mixed dentition. Despite the fact that such treatment could disturb the bone growth of the alveolar process and consequently lead to orthodontic defects [6,10], the occlusal splint may reduce muscle activity and provide greater comfort to patients. However, although occlusal appliances are widely used for treating bruxism in adults, no specific strategy based on appliance therapy has been established for children. Therefore, further studies are needed to investigate the effectiveness of occlusal appliance therapy in children [34]. 

Another therapy proposed for reducing bruxism in children is physiotherapy. The “self-awareness of movement” concept was put forth to improve the organization and coordination of body movements [6]. Nevertheless, it was analyzed only by a few studies, and so its actual impact on bruxism remains unclear. Acupuncture has been successfully used to treat bruxism, resulting in a reduction in the activity of the masseter and anterior temporal muscles, as well as anxiety. Stimulation of specific acupuncture points, with the use of needles (dry needling), infrared radiation, electric current, or laser, can alter the dynamics of circulation and promote muscle relaxation, thereby relieving muscle spasms, inflammation, and pain. In children, acupoint stimulation with laser is recommended because it is painless and requires a shorter exposure time [34]. In addition, photobiomodulation was used in children with Down syndrome in a study by Salgueiro et al. which showed that acupuncture point treatment relieved the symptoms of bruxism in these children and reduced the levels of cortisol [37,38].

Other treatments, such as psychological therapy, are applied to change undesirable habits and reduce stress to lead to a healthier lifestyle. Pharmacological therapy has been used to reduce stress and anxiety and improve the quality and quantity of sleep, while surgical treatment is used to remove airway obstruction [6,16]. Behavioral strategies include biofeedback, relaxation, and improved sleep hygiene. Of these, biofeedback aims to provide patients with immediate information about their behavior, enabling their awareness and reaction, and is used for both AB and SB. Techniques used in biofeedback strategy include EMG feedback from auditory, vibrational, or electrical stimulation and use of devices to wake up the patient from sleep during an SB episode [10]. Sleep hygiene measures include maintaining good ventilation and silence in the bedroom, relaxing before going to bed, and avoiding caffeine before bedtime. These measures aim at reducing the impact of mental stress on SB, but a randomized controlled trial by Lopez et al. with 16 participants revealed that sleep hygiene and relaxation had no effect on SB [39].

Drugs used for treating bruxism include benzodiazepines, anticonvulsants, beta-blockers, serotonergic and dopaminergic drugs, antidepressants, and muscle relaxants [10] and botulinum toxin [40]. In addition, medications such as propranolol, bromocriptine, and amitriptyline have been proposed for the treatment of bruxism, but have not shown to be effective against this condition. Reports indicate that antidepressants such as citalopram, paroxetine, fluoxetine, and venlafaxine induce secondary bruxism, while clonazepam reduces it due to its muscle-relaxing effect. Previous works have highlighted that positive outcomes were achieved with the use of buspirone, but these are short-term studies. Recent reports indicated that administration of hydroxyzine in children for two months improved bruxism, with no adverse effects [36,41]. However, further research is needed to confirm the effects of this drug. Mostafavi et al. investigated the effectiveness of low and moderate doses of diazepam in children with SB, but found that diazepam was not more effective, compared to placebo, in the long-term control of SB in the studied population [35].

### 4.4. Study Limitations

There is no sufficient, statistically relevant data to create recommendation for an everyday clinical practice and a lack of multi-center controlled clinical trials on this issue. However we have focused on the need for education about bruxism in children and further research in this field. 

## 5. Conclusions

SB is an important clinical condition in children, due to the fact that tooth grinding is often very intense and reappears over a long period of time. It causes tooth wear, headaches, facial muscle pain, discomfort during chewing, and limited mouth opening. Bruxism is a nosological entity that should be known to the medical community to ensure its correct identification and treatment. The primary care physician should accurately diagnose bruxism in children, educate parents, prevent potential consequences for oral health, and identify possible comorbidities. In addition, appropriate clinical guidelines to treat SB in children should be developed. The results of the studies considered in this review were clinically relevant and suggested that high sleep hygiene can be helpful in the treatment of SB in children.

## Figures and Tables

**Figure 1 ijerph-18-09544-f001:**
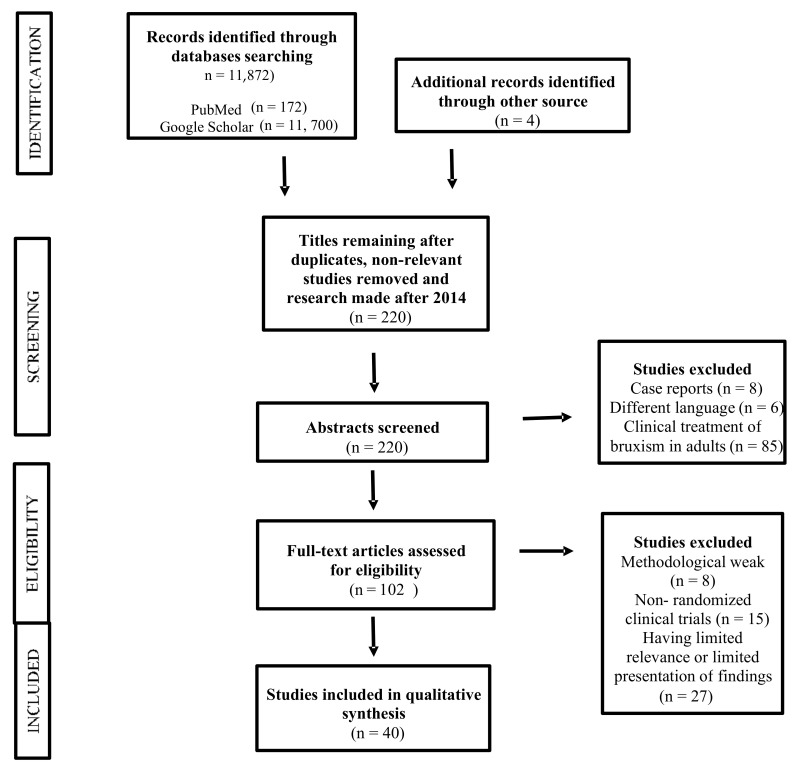
PRISMA flowchart for the systematic review.

**Table 1 ijerph-18-09544-t001:** AASM clinical diagnostic criteria for SB [24].

Patient History	Clinical Evaluation
- Recent patient, parent, or sibling report of the occurrence of tooth-grinding sounds during sleep for at least 3–5 nights per week in the last 3–6 months	- Abnormal tooth wear - Hypertrophy of the masseter muscles on voluntary forceful clenching - Discomfort, fatigue, or pain in the jaw muscles (transient, morning jaw-muscle pain and headache)

**Table 2 ijerph-18-09544-t002:** Methods used for the treatment of SB in children.

Treatment Method	Authors
Use of low and moderate doses of diazepam	S.N. Mostafavi et al. [35]
Oral administration of hydroxyzine	A. Ghanizadeh [36]
Photobiomodulation	M.C.C. Salgueiro et al. [37]
Orthodontic treatment (rapid palatal expansion)	A. Bellerive et al. [33]

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
