# Peer review of "Sleep Bruxism in Children: Etiology, Diagnosis, and Treatment—A Literature Review"

_ijerph, 2021, doi:10.3390/ijerph18189544_

Round 1
Reviewer 1 Report
This is interesting systematic review.
However it has some points which have to be explained:
1. All cited definitions of bruxism have to come from American Academy of Sleep Medicine guidelines, statements etc and Bruxism International Consensus from 2018; Lobbezoo F, Ahlberg J, Raphael KG, Wetselaar P, Glaros AG, Kato T, Santiago V, Winocur E, De Laat A, De Leeuw R, Koyano K, Lavigne GJ, Svensson P, Manfredini D. International consensus on the assessment of bruxism: Report of a work in progress. J Oral Rehabil. 2018 Nov;45(11):837-844. doi: 10.1111/joor.12663. Epub 2018 Jun 21. PMID: 29926505; PMCID: PMC6287494.
Bruxism is not a parafunction anymore!
2. The manuscript content suggest that this is systematic review. Therefore the structure has to be reported in accordance to the PRISMA Statement. Authors have to follow the PRISMA checklist file and revise whole manuscript strictly to the recommendations (from title to the conslusions).
3. The language of the manuscript has to be revised by native speaker.
Author Response
Response to Reviewer No1 Comments
Point 1:
All cited definitions of bruxism have to come from American Academy of Sleep Medicine guidelines, statements etc and Bruxism International Consensus from 2018; Lobbezoo F, Ahlberg J, Raphael KG, Wetselaar P, Glaros AG, Kato T, Santiago V, Winocur E, De Laat A, De Leeuw R, Koyano K, Lavigne GJ, Svensson P, Manfredini D. International consensus on the assessment of bruxism: Report of a work in progress. J Oral Rehabil. 2018 Nov;45(11):837-844. doi: 10.1111/joor.12663. Epub 2018 Jun 21. PMID: 29926505; PMCID: PMC6287494.
Bruxism is not a parafunction anymore!
Response point 1:
References have been added throughout the manuscript as recommended by the Reviewer
Point 2:
The manuscript content suggest that this is systematic review. Therefore the structure has to be reported in accordance to the PRISMA Statement. Authors have to follow the PRISMA checklist file and revise whole manuscript strictly to the recommendations (from title to the conslusions).
Response point 2:
Thank You for Your comments, our review was not prepared following the instruction for a systematic review, but for literature review, therefore we did not follow the protocol and recommendations for PROSPERO and PRISMA statement. In our opinion there is not sufficient scientific data to draw the conclusion for the therapy, as required in systematic review. We have included this information in the manuscript. There is no sufficient, statistically relevant data to create recommendation for a everyday clinical practice and a lack of multi-center controlled clinical trials on this issue. However we have focused on the need for education about bruxism in children.
Point 3:
The language of the manuscript has to be revised by native speaker
Response point 3:
The document was revised and edited by a medical translator Transl Med Poland.

Reviewer 2 Report
The paper is quite interesting, but I have some concerns.
In the introduction, authors write that there are two types of Bruxism (SB and AB), but the review seems to include only sleep bruxism. This should be specified in the title and also in the research methods. About review methodology, did authors PROSPERO registration?
Authors included also articles in Polish, but in this way the research is not reproducible.
Some grammar errors:
- line 58 : no comma is required between "children" and "Determinating"
- line 160: insert reference
- line 238: discussion about adult treatment, no relevant for the review, I suggest to delete it. The same for line 290.
Author Response
Response to reviewer No1 comments:
Point 1:
In the introduction, authors write that there are two types of Bruxism (SB and AB), but the review seems to include only sleep bruxism. This should be specified in the title and also in the research methods.
Response point 1:
In fact, a valuable note, the term bruxism has been made more specific, both in the title and in the methodology and throughout the text of the manuscript, we have corrected the term bruxism on sleep bruxism.
Point 2:
About review methodology, did authors PROSPERO registration?
Response point 2:
Thank You for Your comments, our review was not prepared following the instruction for a systematic review, but for literature review, therefore we did not follow the protocol and recommendations for PROSPERO and PRISMA statement. In our opinion there is not sufficient scientific data to draw the conclusion for the therapy, as required in systematic review. We have included this information in the manuscript. There is no sufficient, statistically relevant data to create recommendation for a everyday clinical practice and a lack of multi-center controlled clinical trials on this issue. However we have concentrated on the problem of bruxism in children.
Point 3:
Authors included also articles in Polish, but in this way the research is not reproducible.
Response point 3:
References included articles in English, but also items in two languages, Polish and English at once
Point 4:
Some grammar errors:
- line 58 : no comma is required between "children" and "Determinating"
- line 160: insert reference
- line 238: discussion about adult treatment, no relevant for the review, I suggest to delete it. The same for line 290.
Response point 4:
In line 58 coma was removed
in line 160 a reference was added
in line 238 and 290 a discussion about adult bruxism was deleted

Round 2
Reviewer 2 Report
Authors correctly answered to previous comments, modifying the manuscript where needed. Therefore, in this version, manuscript can be accepted for publication.